# Alternative Formulation of Antenna Arrays for DF Systems Considering Active-Element Patterns and Scattering Matrices

**DOI:** 10.3390/s21155048

**Published:** 2021-07-26

**Authors:** Bernardo Fabiani, Eduardo Sakomura, Eduardo Silveira, Daniel Nascimento, Daniel Ferreira, Marcelo Pinho

**Affiliations:** Laboratory of Antennas and Propagation, Aeronautics Institute of Technology, Sao Jose dos Campos 12228-900, Brazil; bernardomoscardinifabiani@gmail.com (B.F.); eduardosakomura@gmail.com (E.S.); eduardosilveiranes@gmail.com (E.S.); danielbf@ita.br (D.F.); mpinho@ieee.org (M.P.)

**Keywords:** direction finding, direction-of-arrival, microstrip antenna array, multiple signal classification MUSIC, mutual coupling, polarization diversity, printed monopole array

## Abstract

Direction finding (DF) systems are used to determine the direction-of-arrival (DoA) of electromagnetic waves, thus allowing for the tracking of RF sources. In this paper, we present an alternative formulation of antenna arrays for modeling DF systems. To improve the accuracy of the data provided by the DF systems, the effects of mutual coupling in the array, polarization of the received waves, and impedance mismatches in the RF front-end receiver are all taken into account in the steering vectors of the DoA algorithms. A closed-form expression, which uses scattering parameter data and active-element patterns, is derived to compute the receiver output voltages. Special attention is given to the analysis of wave polarization relative to the DF system orientation. Applying the formulation introduced here, a complete characterization of the received waves is accomplished without the need for system calibration techniques. The validation of the proposed model is carried out by measurements of a 2.2 GHz DF system running a MUSIC algorithm. Tests are performed with a linear array of printed monopoles and with a planar microstrip antenna array having polarization diversity. The experimental results show DoA estimation errors below 6° and correct classification of the polarization of incoming waves, confirming the good performance of the developed formulation.

## 1. Introduction

Direction finding (DF) systems are employed to estimate the direction-of-arrival (DoA) of electromagnetic waves, thus allowing for the tracking of radiofrequency sources in both military and civilian scenarios, such as surveillance, security, navigation, or even rescue [1,2,3]. Among the countless applications of DF systems, we can point out the localization of pirate radios in the vicinity of airports [4], jamming attack detection [5,6], surveillance of borders and restricted areas [7], and security of events [8]. More recently, the use of drones by terrorist organizations has been a concern of the international community [9], and DF systems capable of localizing drone controllers could help police to catch criminals.

Typically, a DF system is composed of an antenna array, a circuit for conditioning the received signals, and a processor running an algorithm that estimates the DoA of the incoming waves. Many antenna array configurations are presented in the literature as solutions for DF systems, as well as a variety of algorithms for DoA estimation [1,2,10,11,12,13,14,15,16]. However, due to formulation complexity, some algorithms for DoA estimation do not often consider the inherent properties of antenna arrays [17], i.e., the radiation patterns, mutual coupling, and polarization are not taken into account in the calculations [15,18,19]. To compensate for this simplification, some calibration techniques have been proposed [20,21,22], but in general, they involve an additional phase of measurements. On the other hand, some improved models consider mutual coupling [13,23,24,25] and antenna polarization [26,27,28] in their formulations to enhance the accuracy of the DoA estimation and to reduce the complexity of the aforementioned calibration stage.

For example, [29] investigates the performance of a DF system through an experimental study that uses mutual coupling compensation, indicating that the DoA estimation can be significantly improved when the not-negligible mutual coupling effect is properly accounted for. The use of the multiple signal classification (MUSIC) algorithm for non-circular signals in the presence of mutual coupling is treated in [30], which derives closed-form expressions for the biases in uniform linear arrays. In [31], a 2D DoA estimation algorithm is presented, based on MUSIC, for the uniform rectangular array (URA) in the presence of mutual coupling.

The treatment of the wave polarization can be found, for example, in [26], which proposes a low-angle tracking algorithm that uses data collected from a polarization-sensitive array for very-high frequency radars. In [27], a mathematical model of a dual-polarized circular array is reported as well as the analysis of a joint estimation method, developed from the MUSIC algorithm, that provides the DoA and polarization parameters of radar signals. The polynomial-rooting-based algorithm for DoA and polarization estimation is, in turn, applied in [32] using data measured with antennas positioned arbitrarily in space. In [33], the improvement in the performance of the MUSIC algorithm is accomplished by taking into account the theoretically derived radiation patterns of crossed-loop monopole antennas used in an array for the measurement of ocean surface targets. Experimental results demonstrate the performance enhancement.

As manifested by the previous brief literature review, mutual coupling, radiation pattern, and polarization in DF systems are very promising research topics, mainly due to the improvement they bring to the characterization of the incoming waves. In this context, this paper presents an alternative formulation of antenna arrays that does not require calibration of the DF system and takes into account mutual coupling, radiation pattern, and wave polarization. In addition, the proposed formulation is structured to compute the transfer function of the RF front-end receiver, resulting in a closed-form expression that relates the available complex voltages at the receiver outputs to the incident electric field on the array. The derived expression uses the scattering matrices of both the array and the receiver as well as the active-element patterns of the antennas in the array. The formulation can be directly applied to construct the steering vectors used in the DoA algorithms. To the best of our knowledge, this is the first work to present such a formulation using scattering matrices and active-element patterns to build the transfer function of DF systems. The comparison between the characteristics of the proposed method and those of the other techniques discussed here is synthesized in Table 1, which lists the antenna parameters considered in the formulation, the capability to estimate the polarization of incoming waves, and the need for calibration.

The validation of the developed formulation is carried out with tests in an anechoic chamber using a 2.2 GHz DF system designed and built on the premises of the Laboratory of Antennas and Propagation at the Aeronautics Institute of Technology. All parts of the DF system are detailed in the following sections with emphasis on the receiving circuitry based on IQ demodulator. Moreover, for accurate characterization of RF sources, special attention is given to the description of the polarization of the incoming waves, considering the orientation of the receiving antenna array. This analysis is of great utility for DF systems installed on moving platforms [34]. By using prototypes of printed arrays, two distinct situations are studied: in the first one, a linear array of six quarter-wave monopoles is considered, since this type of array is widely used in DF systems [15,20,21], whilst in the second one, a planar microstrip antenna array with polarization diversity is employed, thus allowing for the identification of any wave polarization state. In the experimental tests, the polarization and direction of the incoming waves were very well estimated with exceedingly small errors (less than 6°). Additionally, it is important to mention that since the transfer function of the receiving antenna array (obtained from electromagnetic simulations) and that of the receiver circuit (obtained from bench measurements with a vector network analyzer—VNA) are carefully considered in our model, the DF system does not require any type of calibration.

After this introduction, the paper is divided into the following sections. Section 2 presents the derivation of the transfer function of the DF system, which gives the output voltages of the front-end circuit as a function of the incident electric field and takes into account each antenna’s radiation pattern, polarization, impedance mismatches, and mutual coupling. In Section 3, the derived transfer function is used to compute the steering vectors of the MUSIC algorithm. The design, construction, and tests of each hardware component of the 2.2 GHz DF system are described in Section 4. In the penultimate section of the paper, the system integration and tests in an anechoic chamber are illustrated. Finally, some comments on the proposed formulation are summarized in Section 6.

## 2. DF System and Mathematical Modeling

The architecture of the reference DF system consists of five main parts: antenna array with *N* elements (not necessarily equal and positioned in a regular geometrical arrangement); multiplexer circuit; comparator or conditioning signal circuit; data acquisition; and processor board running the DoA algorithm, as illustrated in Figure 1. The multiplexer circuit is responsible for switching one antenna at a time to the comparator, which, in turn, compares the selected signal to a reference one (local oscillator—LO). Since all available signals at the antennas’ terminals are compared to the LO, these data are collected in the data acquisition stage, and the relative levels (magnitude and phase) between them are processed by the algorithm that estimates the DoA.

In order to obtain more accurate results with the DF algorithm, initially, a transfer function of the system (Figure 1) must be established. The formulation is derived independently of the circuit components and antenna array, which makes it general and applicable to other types of DF system architectures. Considering the mutual coupling in the array and the impedance mismatches between all components, a transfer function based on the *S* parameters of cascaded components is determined. The main idea is to derive an expression that relates the complex voltage ($V_{S}$) at the input of the comparator circuit to the complex electric field $\left({\vec{E}}_{w}\right)$ of the plane wave incident on the antennas. This is accomplished by solving the system of cascaded S matrices that relate the voltage waves defined in Figure 1, as will be described below.

### 2.1. Antenna Array Modeling

The first step of the proposed formulation consists of calculating the voltages induced at the antennas’ terminals in terms of the complex incident electric field ${\vec{E}}_{w}$ (written in spherical coordinates relative to the array frame). This could be carried out by numerical simulations [23,35] in which uniform plane waves impinge upon the array and the corresponding voltages induced at the antennas’ terminals are computed, but this process involves a high computational cost. For example, if we consider a step of five degrees and two orthogonal components ($E_{w\theta },E_{w\phi }$), an impressive number of 5044 simulations is required to determine a complete transfer function, which is probably not feasible in most situations. To overcome this problem, an alternative post-processing formulation is proposed here. When this approach is used, both incident wave polarization and mutual coupling in the array are incorporated into the model to provide more accurate results.

Initially, we evaluate the voltages induced at the antennas’ terminals by assuming that the antennas are terminated in matched loads. To do this, we start with the well-known result for the voltage induced at the terminals of an open-circuited radiator due to an incident electric field ${\vec{E}}_{w}$ [17]. For the nth array element, the phasor of this voltage can be written as
(1)$$V_{oc}^{n}=c\langle {\vec{E}}_{w}|{\vec{E}}_{oc}^{n}\left(\theta_{w},\phi_{w}\right)\rangle ,$$
where n=1, 2, ⋯, N, c is a proportionality constant, and ${\vec{E}}_{oc}^{n}\left(\theta_{w},\phi_{w}\right)$ is the complex electric field radiated by the *n*th element in the direction $\left(\theta_{w},\phi_{w}\right)$ of the incoming plane wave when it is driven with an ideal current source and all other elements are open-circuited. The angle $\theta_{w}$ is measured between the z-axis and the propagation vector of the plane wave, whereas $\phi_{w}$ is the angle between the x-axis and the projection of the propagation vector on the xy-plane. The complex field ${\vec{E}}_{oc}^{n}$ is expressed in spherical coordinates relative to the array frame, and the frequency f of the ideal current source is the same as that of the incident electric field. The symbol 〈·|·〉 denotes the inner product, and ${\vec{E}}_{w}$ is evaluated at the coordinate origin.

Now, if all radiators are terminated with matched loads $Z_{o}$, then the voltages induced at the antennas’ terminals due to the incident electric field ${\vec{E}}_{w}$ can be computed as
(2)$$\left[V_{a}^{-}\right]={\left(\left[I\right]+\left(\left[I\right]+\left[S^{a}\right]\right){\left(\left[I\right]-\left[S^{a}\right]\right)}^{-1}\right)}^{-1}\left[V_{oc}\right],$$
in which [I] is an identity matrix of order *N*, $\left[S^{a}\right]$ is the scattering matrix of the antenna array at the frequency f of the incoming wave, and $\left[V_{oc}\right]$ is an N×1 column matrix whose elements are the open-circuit voltages given by (1). Note that for matched systems, the magnitudes of the voltages in $\left[V_{a}^{-}\right]$ are the same as those of the voltage waves that propagate in the lossless transmission lines of characteristic impedance $Z_{o}$ connected at the terminals of the antennas.

Additionally, commercial software packages, such as Ansys HFSS, usually provide the active-element pattern [36] of each array element (i.e., the radiated electric field obtained by exciting the one element with an incident voltage wave of unit amplitude and terminating all others with matched loads). Notice that only one full-wave simulation is required to evaluate all active-element patterns of an array. In this way, it is appropriate to rewrite $\left[V_{oc}\right]$ in (2) in terms of the column matrix $\left[{\vec{E}}_{at}\left(\theta_{w},\phi_{w}\right)\right]$ of the active-element patterns $\vec{E_{at}^{n}}\left(\theta_{w},\phi_{w}\right)$ computed at the direction $\left(\theta_{w},\phi_{w}\right)$ and frequency f. As a consequence, the voltages induced at the terminals of a matched antenna array are then given by
(3)$$\left[V_{a}^{-}\right]=c\left[L\right]\;h\left(\left[{\vec{E}}_{at}\left(\theta_{w},\phi_{w}\right)\right]\right),$$
where
(4)$$\left[L\right]={\left(\left[I\right]+\left(\left[I\right]+\left[S^{a}\right]\right){\left(\left[I\right]-\left[S^{a}\right]\right)}^{-1}\;\right)}^{-1}\;{\left(\left[I\right]-\left[S^{a}\right]\right)}^{-1}$$
and $h\left(\cdot \right)=\langle {\vec{E}}_{w}|\cdot \rangle$.

An array of six printed quarter-wave monopoles spaced a half-wavelength apart, which will be discussed in Section 4.1.1, was analyzed in Ansys HFSS software to test Equation (3). First, the array was simulated as a transmitting antenna to extract the scattering matrix ($\left[S^{a}\right]$) and the active-element patterns ($\left[{\vec{E}}_{at}\right]$) required by (3). Next, three other simulations were carried out to analyze the array in the receiving mode. Specifically, we determined the voltages induced at the terminals of the six antennas when linearly polarized uniform plane waves propagating in directions in the *xy*-plane $\left(\theta_{w}=90{}^{\circ}\right)$ were incident on the array. These voltages were also computed with the aid of (3) and they are compared in Table 2 for the three angles of incidence considered, i.e., $\phi_{w}=30{}^{\circ}$, $\phi_{w}=60{}^{\circ}$, and $\phi_{w}=90{}^{\circ}$ (direction of the array axis), and with the incident electric field in the θ-direction (i.e., ${\vec{E}}_{w}=1\hat{\theta }$ V/m). The results show an excellent match, thus proving that the proposed formulation can properly estimate the voltages at the antennas’ terminals and with the advantage of being computationally efficient since it requires only one full-wave simulation per frequency (to calculate both $\left[{\vec{E}}_{at}\right]$ and $\left[S^{a}\right]$). The small differences observed in the comparisons of the data in Table 2 are due to the influence of the mesh at the antennas’ ports on the line integral used to evaluate the voltages.

As the computation of (3) is extremely fast, the active-element patterns can be tabulated using small steps of 0.5°—for example, for both spherical coordinate angles θ and ϕ—which contributes to the improvement of the resolution of the DoA algorithm. Furthermore, knowledge of the proportionality constant c is not required because DoA algorithms deal with the relative voltages between the antennas.

### 2.2. Receiving Circuit Modeling

In order to adequately estimate the voltage $V_{S}^{-}$ incident at the input of the comparator device, a transfer function of the receiving circuit must be determined and integrated with the antenna array model discussed previously.

The receiving circuit model can be derived with reference to Figure 1, in which an array of N antennas is employed. In this configuration, the scattering matrix of the multiplexer circuit, evaluated at the frequency f and of order N+1, has the following form
(5)$$\left[S^{Mult}\right]=\left[\begin{array}{cc} {\left[S_{in}^{Mult}\right]}_{N\times N} & {\left[rev\right]}_{N\times 1}\\ {\left[gain\right]}_{1\times N} & {\left[S_{out}^{Mult}\right]}_{1\times 1} \end{array}\right],$$
which also accounts for the transfer functions (insertion losses and phases) of cables and connectors attached to the multiplexer circuit.

In (5), the row submatrix [gain] contains the transmission coefficients between the input and output ports of the multiplexer circuit, whereas the column submatrix [rev] consists of the reverse transfer functions from the output to the input ports. The submatrix $\left[S_{out}^{Mult}\right]$, in turn, is the reflection coefficient at the output port (N+1) itself when all input ports are matched, and the submatrix $\left[S_{in}^{Mult}\right]$ contains the reflection coefficients at the input ports (1 to N) as well as the coupling coefficients between them for the output port terminated in a matched load.

Since the multiplexer circuit switches one of the N inputs to a common output (port N+1), N different matrices $\left[S^{Mult}\right]$ are required to properly describe the receiving circuit at a given frequency. Depending on the design choice, the input ports of the MUX not switched to the output can be internally terminated to a matched load (which is referred to as non-reflective configuration) or left open-circuited (which is known as reflective configuration). The formulation presented here can handle both cases.

If the reflection coefficient at the comparator input port is denoted as $\left[S^{c}\right]$, we can write the following system of matrix equations to relate the voltage waves in the circuit of Figure 1:(6)$$\{\begin{array}{c} \begin{array}{c} \left[V_{n}^{-}\right]=\left[S^{a}\right]\;\left[V_{n}^{+}\right]+\left[V_{a}^{-}\right] \end{array}\\ \begin{array}{c} \left[\begin{array}{c} \left[V_{n}^{+}\right]\\ \left[V_{S}^{-}\right] \end{array}\right]=\left[S^{Mult}\right]\left[\begin{array}{c} \left[V_{n}^{-}\right]\\ \left[V_{S}^{+}\right] \end{array}\right] \end{array}\\ \left[V_{S}^{+}\right]=\left[S^{c}\right]\;\left[V_{S}^{-}\right] \end{array}$$
where the column matrix $\left[V_{a}^{-}\right]$ is given by (3) and contains the voltages at the antennas’ terminals, assuming they are perfectly matched, and the column matrices $\left[V_{n}^{-}\right]$ and $\left[V_{n}^{+}\right]$ contain, respectively, the forward and backward voltage waves at the input of the transmission lines that connect the antennas to the multiplexer circuit.

Solving the linear system in (6) for the voltage wave $\left[V_{S}^{-}\right]$ yields
(7)$$\left[V_{S}^{-}\right]=\left({\left(\;\left[I\right]-\left[U\right]\;\left[S^{a}\right]\;\left[rev\right]\;\left[S^{c}\right]\;\right)}^{-1}\left[U\right]\right)\;\left[V_{a}^{-}\right]$$
with
(8)$$\left[U\right]={\left(\;\left[I\right]-\left[S_{out}^{Mult}\right]\;\left[S^{c}\right]\;\right)}^{-1}\left[gain\right]\;{\left(\;{\left[I\right]}_{n\times n}-\left[S^{a}\right]\;\left[S_{in}^{Mult}\right]\;\right)}^{-1}.$$

Finally, note that the receiving system is fully characterized by the above equations, which take into account the mutual coupling in the array and the imperfections of the MUX circuit (e.g., mismatches and coupling between ports). Notice also that each MUX state requires the evaluation of one transfer function (7) in order to characterize the voltages produced by each antenna of the array. The flowchart presented in Figure 2 summarizes the steps required to compute the voltages at the receiver output, which are used in the DoA algorithms, as will be discussed in the next section.

## 3. DoA Algorithm Implementation

By making use of (7) together with the MUSIC algorithm, a computer program is written in MATLAB, allowing for the estimation of the DoA and polarization of incoming electromagnetic waves. The choice for MUSIC is mainly motivated by its widespread use by the scientific community [1,2,14,15,18,19,28,30,32], which provides a vast number of references that detail the algorithm potentials and characteristics of its implementation.

### 3.1. MUSIC Algorithm

The MUSIC algorithm implemented in this section is based on the developments presented in [37,38], which also describe flowcharts summarizing the algorithm pseudocode. Initially, we consider the possibility of polarization diversity, so that two steering vectors, denoted as $S_{i\theta }\left(\theta ,\phi \right)$ and $S_{i\phi }\left(\theta ,\phi \right)$, must be filled with the aid of (7). These vectors contain the voltages that appear at the comparator input for each MUX state when an incident electric field $\vec{E}$ arrives from (θ,ϕ) and is decomposed into their orthogonal components $E_{\theta }$ and $E_{\phi }$, respectively. The spherical coordinate angles θ and ϕ may cover all space (i.e., 0°≤θ≤180° and 0°≤ϕ<360° with steps as small as desired to meet the required resolution) or only a specific region where the DF system is intended to scan. As a result, there is a pair of steering vectors for each direction (θ,ϕ).

More specifically, $S_{i\theta }{\left(\theta ,\phi \right)}^{T}=\left[V_{S_{1}}^{-}\;V_{S_{2}}^{-}\;\ldots \;V_{S_{N}}^{-}\right]$ (the superscript T represents the transpose) is a vector containing *N* values of $V_{S}^{-}$, one for each of the N states of the MUX circuit, considering an incident electric field $\vec{E}=E_{\theta }\hat{\theta }$ arriving at (θ,ϕ). Similarly, $S_{i\phi }\left(\theta ,\phi \right)$ contains the same N voltages, but for $\vec{E}=E_{\phi }\hat{\phi }$. Without loss of generality, we can assume $E_{\theta }$ and $E_{\phi }$ are equal to 1 V/m in both cases.

Once the steering vectors have been computed, an auxiliary matrix $S_{\mathrm{aux}}\left(\theta ,\phi \right)$ is found as
(9)$$S_{\mathrm{aux}}\left(\theta ,\phi \right)=\left[\begin{array}{c} S_{i\theta }^{H}\left(\theta ,\phi \right)\\ S_{i\phi }^{H}\left(\theta ,\phi \right) \end{array}\right]U_{L}U_{L}^{H}\left[\begin{array}{cc} S_{i\theta }\left(\theta ,\phi \right) & S_{i\phi }\left(\theta ,\phi \right) \end{array}\right],$$
in which the superscript H denotes the Hermitian transpose (i.e., the conjugate transpose), and the matrix $U_{L}$, of dimension N×N−p, is referred to as the noise subspace matrix, with p being the number of incident waves on the array [37]. The columns of $U_{L}$ are the N−p eigenvectors belonging to the lower eigenvalues of the following sample spatial covariance matrix:(10)$$\hat{C}=\frac{1}{M}\sum_{m=1}^{M}Y_{m}Y_{m}^{H},$$
where $Y_{m}^{T}=\left[Y_{m1}\;Y_{m2}\;\cdots \;Y_{mN}\right]$, m∈{1,…,M}, is the mth term of the sequence of M vectors containing the N voltages measured by the comparator prototype, one for each state of the MUX circuit, and produced by the plane wave incident on the antenna array. In order to guarantee the accuracy of the results provided by the MUSIC algorithm, the DoA and polarization of the incoming waves must remain unchanged during the time the M samples are collected.

The DoA estimation is performed through the $S_{\mathrm{MUSIC}}$ function, which is evaluated as
(11)$$S_{\mathrm{MUSIC}}\left(\theta ,\phi \right)={\left(\lambda_{\min}\left(S_{\mathrm{aux}}\left(\theta ,\phi \right)\right)\right)}^{-1},$$
where the operator $\lambda_{\min}\left(\cdot \right)$ returns the lowest eigenvalue of the matrix in its argument. Since $S_{\mathrm{aux}}\left(\theta ,\phi \right)$ (9) is a square matrix of order 2, its lowest eigenvalue is easily found as
(12)$$\lambda_{\min}=\left(s_{11}+s_{22}-\sqrt{{\left(s_{11}-s_{22}\right)}^{2}+4s_{12}s_{21}}\right)/2,$$
with $s_{uv}$ denoting the element in the uth row and vth column of $S_{\mathrm{aux}}$.

Scanning the space (θ,ϕ), the peaks of the function $S_{\mathrm{MUSIC}}$ (11) occur at the directions $\left(\theta_{w},\phi_{w}\right)$ of the incoming waves. In addition, the eigenvector of the matrix $S_{\mathrm{aux}}\left(\theta_{w},\phi_{w}\right)$ belonging to the lowest eigenvalue $\lambda_{\min}$ can be written in the form ${\left[\begin{array}{cc} 1 & q \end{array}\right]}^{T}$ [37], where the value of q gives the complex ratio between the orthogonal components of the incident electric field ${\vec{E}}_{w}$, i.e., $q=E_{w\phi }/E_{w\theta }$. Thus, both the direction and polarization of the incident wave are accurately estimated, since by using the proposed formulation, the steering vectors in the MUSIC algorithm take into account the mutual coupling in the array, the radiation pattern of the antennas, and the impedance mismatches in the receiving circuit.

Note, however, that depending on the chosen array, the polarization of the incident plane wave cannot be computed, and the DoA estimation is ambiguous: for example, linear arrays of equally spaced electric monopoles or electric dipoles. Therefore, the array must be carefully designed to allow the determination of an ambiguous DoA and the polarization of the incoming waves.

### 3.2. Coordinate System Treatment

As stated in Section 2, both the incident electric field ${\vec{E}}_{w}$ and the active-element patterns $\vec{E_{at}^{n}}$ are expressed in the coordinate system attached to the antenna array. Hence, the DoA $\left(\theta_{w},\phi_{w}\right)$ and the ratio $q=E_{w\phi }\left(\theta_{w},\phi_{w}\right)/E_{w\theta }\left(\theta_{w},\phi_{w}\right)$ computed with the MUSIC algorithm are relative to this frame of reference. However, if the DF system is installed on moving platforms (e.g., drones, airplanes), it is useful to express the DoA and the ratio between the components of the incident electric field in a coordinate system fixed on the ground, i.e., the direction $\left(\theta_{w}^{’},\phi_{w}^{’}\right)$ and the ratio q’. For example, when the DF system is mounted on a drone and tracks a ground-based radar, it is often more convenient to monitor the DoA and the ratio between the components regardless of the drone orientation in space.

Since the moving platforms are usually equipped with an inertial measurement unit, the angles between these two coordinate systems are known, and the relationships between $\left(\theta_{w},\phi_{w}\right)$ and $\left(\theta_{w}^{’},\phi_{w}^{’}\right)$ as well as q and q’ can be promptly evaluated. Figure 3 shows the rectangular coordinate systems attached to the antenna array (xyz) and to the ground (x’y’z’), from which the spherical coordinate angles $\theta_{w},\;\phi_{w}$ and $\theta_{w}^{’},\;\phi_{w}^{’}$ are measured, respectively. The angle α depicted in the figure is the angle between the z-axis and the y’z’-plane and is contained in the xz-plane. The angle β is, in turn, the angle between the xz-plane and the z’-axis and is contained in the y’z’-plane. The positive values of α and β follow the orientations indicated in Figure 3.

The unit vectors $\hat{x}’$, $\hat{y}’$, and $\hat{z}’$ in the rectangular coordinate system attached to the ground can be transformed to the unit vectors $\hat{x}$, $\hat{y}$, and $\hat{z}$ in the rectangular coordinate system attached to the array using the following linear transformation [39]:(13)$$\left[\begin{array}{c} \hat{x}\\ \hat{y}\\ \hat{z} \end{array}\right]=P\left[\begin{array}{c} \hat{x}’\\ \hat{y}’\\ \hat{z}’ \end{array}\right],$$
where
(14)$$P=\left[\begin{array}{ccc} \cos\alpha & -\sin\alpha \sin\beta & -\sin\alpha \cos\beta \\ 0 & \cos\beta & -\sin\beta \\ \sin\alpha & \cos\alpha \sin\beta & \cos\alpha \cos\beta \end{array}\right].$$

From Figure 3, the radial unit vectors $\hat{r}$ and $\hat{r}’$ are equal; then, the spherical coordinate angles $\theta_{w}^{’}$ and $\;\phi_{w}^{’}$ can be determined through the relations
(15)$$\cos\theta_{w}^{’}={\hat{z}}^{’}\cdot {\hat{r}}^{’}={\hat{z}}^{’}\cdot \hat{r}=p_{t31}\sin\theta_{w}\cos\phi_{w}+p_{t32}\sin\theta_{w}\sin\phi_{w}+p_{t33}\cos\theta_{w},$$
(16)$$\tan\phi_{w}^{’}=\frac{{\hat{y}}^{’}\cdot {\hat{r}}^{’}}{{\hat{x}}^{’}\cdot {\hat{r}}^{’}}=\frac{{\hat{y}}^{’}\cdot \hat{r}}{{\hat{x}}^{’}\cdot \hat{r}}=\frac{p_{t21}\sin\theta_{w}\cos\phi_{w}+p_{t22}\sin\theta_{w}\sin\phi_{w}+p_{t23}\cos\theta_{w}}{p_{t11}\sin\theta_{w}\cos\phi_{w}+p_{t13}\cos\theta_{w}},$$
with $p_{tuv}$ denoting the element in the uth row and vth column of the transpose of P (which is the inverse of P itself).

Now, to determine the ratio q’ at a direction (θ’,ϕ’) in the ground coordinate system (which may not necessarily coincide with $\left(\theta_{w}^{’},\phi_{w}^{’}\right)$), the corresponding unit vectors $\hat{\theta }’$ and $\hat{\phi }’$ are related to the unit vectors ${\hat{\theta }}_{w}$ and ${\hat{\phi }}_{w}$ at the direction $\left(\theta_{w},\phi_{w}\right)$ in the array coordinate system in the following way:(17)$$\left[\begin{array}{c} {\hat{\theta }}_{w}\\ {\hat{\phi }}_{w} \end{array}\right]=R\left[\begin{array}{c} \hat{\theta }’\\ \hat{\phi }’ \end{array}\right],$$
where
(18)$$R=\left[\begin{array}{ccc} \cos\theta_{w}\cos\phi_{w} & \cos\theta_{w}\sin\phi_{w} & -\sin\theta_{w}\\ -\sin\phi_{w} & \cos\phi_{w} & 0 \end{array}\right]P\left[\begin{array}{cc} \cos\theta ’\cos\phi ’ & -\sin\phi ’\\ \cos\theta ’\sin\phi ’ & \cos\phi ’\\ -\sin\theta ’ & 0 \end{array}\right].$$

Consequently, the ratio $q^{\prime }=E_{w\phi ’}\left(\theta ’,\phi ’\right)/E_{w\theta ’}\left(\theta ’,\phi ’\right)$ at the direction (θ’,ϕ’) is given by
(19)$$q^{\prime }=\frac{r_{22}q+r_{12}}{r_{21}q+r_{11}}.$$

The procedure discussed in this section can also be applied to situations where the three Euler angles that transform the ground coordinate system to the array coordinate system are known. In this case, the matrix P is replaced by the product of the three rotation matrices associated with the Euler angles.

## 4. Prototypes

A complete 2.2 GHz DF system running the MUSIC algorithm was fabricated and tested to validate the developments set forth thus far. In order to perform three-dimensional DoA estimation, the system was designed to work with four antennas, as will be detailed below. The next sections address the characteristics of each stage of the system. A PNA-L N5230A vector network analyzer from Agilent Technologies was used to measure the scattering parameters of the stages.

### 4.1. Antenna Arrays

This work employs two different antenna arrays: the first one consists of a microstrip antenna array designed to exploit polarization diversity in the DF system, while the second one is a printed monopole array, which enables the validation of the system with a traditional array geometry used in many DF systems.

#### 4.1.1. Printed Monopole Array

Despite not allowing a three-dimensional estimation, a linear array of printed monopoles (Figure 4) is useful for validating our formulation since it is a traditional array geometry usually found in DF systems [15,20,21]. This configuration only allows a two-dimensional scan on the *xy*-plane. In the designed array, the monopoles with dimensions *h* = 34.1 mm and *w* = 10.0 mm are printed on both sides of a 1.6 mm FR4 substrate (*ε_r_* = 4.2 and tan *δ* = 0.02) and positioned above a rectangular ground plane of dimensions 54 cm × 55 cm. The radiators are spaced from each other by 68 mm (λ_0_/2) and are fed through SMA connectors fixed on the ground plane.

This six-radiator array was previously designed to be part of a phased array system based on a microwave beamforming circuit under research in the Laboratory of Antennas and Propagation at the Aeronautics Institute of Technology. For this work, only the four central elements (1 to 4) are employed in the DoA system, while the two external elements (5 and 6) are terminated in matched loads (50 Ω), behaving as parasitic radiators. Figure 5 shows the comparison between the simulated (Ansys HFSS) and measured S parameters of this array. As seen, both simulated and measured reflection $S_{nn}$ and coupling $S_{mn}$ parameters agree very well. For the four monopoles, the reflection parameter is below −12 dB over the range from 2.1 to 2.3 GHz, and the coupling parameters are lower than −15 dB.

#### 4.1.2. Microstrip Antenna Array

Considering a single incident plane wave of unknown polarization, the array must have at least four radiators with distinct polarizations to solve a three-dimensional DoA problem [40]. In this case, even if one of the antennas and the wavefront are cross-polarized (i.e., there is a complete polarization mismatch between them), the other antennas in the array will exhibit nonzero output power, making possible the DoA estimation. For the purpose of this paper, a planar microstrip antenna array with four rectangular linearly polarized (LP) elements is proposed to provide coverage in a cone of aperture 50°. Complete polarization diversity is achieved by arranging each antenna in different relative angles with no matches between their polarizations.

The antenna array prototype was printed on a 3.175 mm thick CuClad 217 laminate (*ε_r_* = 2.2 and tan *δ* = 0.001) with sides of length 250 mm [40]. The dimensions of the radiators were initially established by the cavity model [41] and next optimized in HFSS software, resulting in the following dimensions according to Figure 6a: *W* = 55.65 mm, *L* = 42.80 mm, and *p* = 12.80 mm for the SMA probe position (all radiators are identical). The rotations of antennas 2, 3, and 4, relative to antenna 1, are 140°, 300°, and 200° counterclockwise, respectively. Figure 6b shows a photo of the fabricated prototype, and Figure 7 illustrates the comparison between the simulated and measured magnitude of S parameters, indicating that the prototype works properly at 2.2 GHz. It is important to mention that although the isolation between the antennas is greater than 16 dB at 2.2 GHz, which would be neglected by many MUSIC implementations, it is considered in the formulation proposed here to enhance the quality of the DoA and polarization estimations.

### 4.2. Receiving Circuit

Considering the proposed DF system (Figure 1), the amplitude and phase of the voltages $V_{S_{n}}^{-}$ are digitized for each state of the MUX circuit to allow for running the MUSIC algorithm. It is important to mention that the values of $V_{S_{n}}^{-}$ do not need to be absolute voltages, but instead can be relative to a common reference (LO in this work).

For a system composed of four antennas, the designed receiving circuit has three main parts, as described below.

(1) Selection stage: the MUX circuit selects the signal from an antenna to be compared to the LO in the next stage.

(2) Comparison stage: this circuit compares the signals selected in the prior stage to the LO.

(3) Data acquisition stage: this circuit digitizes the two analog voltages associated with the amplitude and phase determined in the comparison stage.

Next, these three stages are detailed with emphasis on their design and on the transfer function that must be used in (7).

#### 4.2.1. Selection Stage

The proposed selection stage (Figure 8) is a device of five ports, four inputs (numbered from 1 to 4), and one output (number 5), in which only one input is switched to the output at a time. This circuit is comprised of a parallel arrangement of non-reflective, single pole, double throw switches HMC284A [42] from Analog Devices, denoted as MUX 4:1 in Figure 8, and four low-noise amplifiers (LNA) HMC286E (gain of 20 dB and noise figure of 1.9 dB at 2.2 GHz) [43], also from Analog Devices, that minimize the noise figure of the receiver. The circuit is printed on a 14.7 mil-thick CuClad 250GX laminate (*ε_r_* = 2.55 and tan *δ* = 0.002).

By using a vector network analyzer (VNA), the five-port circuit was tested for the four possible states. The measured magnitudes of the transmission coefficients $S_{5n}$ (*n* = 1, …, 4) and reflection coefficients $S_{jj}$ (*j* = 1, …, 5) at each operating state are shown in Figure 9 and Figure 10, respectively. As seen from these figures, the magnitudes of the four $S_{5n}$ are in the interval from 17 dB to 19 dB at 2.2 GHz in the ON states, whereas the magnitudes of $S_{5n}$ are below −10 dB at 2.2 GHz in the OFF states. In turn, the magnitudes of the reflection coefficients $S_{jj}$ are lower than −10 dB at 2.2 GHz for both ON and OFF states, showing good impedance matching at the five ports. Note, however, that the responses from each input port to the output port are not exactly the same. However, this behavior is taken into account in the formulation proposed here since it employs the measured S-matrix of the receiving circuit to express the steering vectors used in the MUSIC algorithm. In addition, the reverse parameters ($S_{n5}$) are on the order of −30 dB at 2.2 GHz, while the isolation between adjacent input ports is about 20 dB at this frequency. Although these quantities do not substantially affect the accuracy of the DoA estimation, they are also considered in the system model because they are part of the scattering matrix $\left[S^{Mult}\right]$.

#### 4.2.2. Comparison and Data Acquisition Stages

As mentioned, the output voltage $V_{S}^{-}$ must be compared to the LO voltage for each of the four states of the MUX. Here, an IQ-demodulator is employed to realize the comparison stage [44]. According to the schematic in Figure 11, the demodulator has RF and LO inputs, denoted as RF*_in_* and LO*_in_*, respectively, and it provides two intermediate-frequency IF output voltages, which pass through low-pass filters LPF. The acquisition stage is, in turn, composed of IF amplifiers and analog-to-digital converters ADC whose output voltages are read by a microcontroller. The architecture of this microwave comparator can be implemented using commercial integrated circuits, and it exhibits high sensitivity and small errors in amplitude and phase detection, as will be discussed next.

For proper operation, LO*_in_* is synchronized to the RF*_in_* frequency (i.e., *f*_RF_ = *f*_LO_). As a consequence, the voltages $V_{I}$ and $V_{Q}$ at the output of the low-pass filters consist of DC levels, which can be written, in general, as
(20)$$V_{I}=A\cos\left(\phi_{\mathrm{RF}}-\phi_{\mathrm{LO}}\right),$$
(21)$$V_{Q}=A\sin\left(\phi_{\mathrm{RF}}-\phi_{\mathrm{LO}}\right),$$
where *A* is an amplitude dependent on the amplitudes of the RF and LO input voltages, and $\phi_{\mathrm{RF}}$ and $\phi_{\mathrm{LO}}$ are the phases of the RF and LO input voltages, respectively.

The phasor that expresses the relative amplitude and phase of the RF voltage with respect to the LO voltage has a magnitude and a phase that can be easily estimated by applying simple trigonometric relations to $V_{I}$ and $V_{Q}$. Then,
(22)$$Magnitude=c’\sqrt{V_{I}{}^{2}+V_{Q}{}^{2}},$$
(23)$$Phase=\arctan2\left[\frac{V_{Q}}{V_{I}}\right],$$
in which c’ is a proportionality constant and the function arctan2 returns the four-quadrant inverse tangent of $V_{Q}$ and $V_{I}$. For some commercially available IQ-demodulators, the voltages $V_{I}$ and $V_{Q}$ have different amplitudes in addition to DC offsets. In such cases, the microwave comparator must be previously calibrated before its operation in order to determine these amplitudes and DC offsets [45]. Expressions similar to (22) and (23) can also be derived to estimate the relative amplitude and phase of the RF voltage with this assumption.

To comply with the system requirements, a comparator based on the ADL5380 IQ-demodulator from Analog Devices was mounted on a 14.7 mil-thick substrate (*ε_r_* = 2.55 and tan *δ* = 0.002), as shown in Figure 12a. This component can operate from 400 MHz to 6 GHz and its error vector magnitude (EVM) is about −20 dB for RF levels of −70 dBm [46], which makes the selected component a good alternative for the comparison stage. Two RC low-pass filters with a cutoff frequency of 480 MHz are connected at the IF outputs of the demodulator to ensure that only DC voltages are transmitted to the IF amplifiers.

The voltages $V_{I}$ and $V_{Q}$ are amplified and digitized by an ADS1115 analog-to-digital converter from Texas Instruments, which is mounted on the backside of the RF circuit (Figure 12b). This ADC can digitize two differential channels with a resolution of 16 bits, and it has an internal programmable gain amplifier, which allows for accurate comparisons of small and large voltages.

Before the integration of the comparison and acquisition stages into the receiving circuit, the reflection coefficient at the RF input port (RF*_in_*) was measured, and its magnitude is presented in Figure 13 for different LO power levels. As seen from this figure, the reflection coefficient at the RF input port is better than −11 dB in the range of 2.0–2.4 GHz for LO power levels between −6 and −2 dBm. In addition, a 2.2 GHz sinusoidal signal with a fixed level of −30 dBm and phase varying from 0° to 360° in steps of 10° was applied to the RF input port. The DC voltages $V_{I}$ and $V_{Q}$ were measured at each step and then plotted versus the phase of the RF input signal. From these plots, we have confirmed that $V_{I}$ and $V_{Q}$ have the same amplitude; however, they also present small DC offsets. Consequently, these DC offsets must be subtracted from the measured values of $V_{I}$ and $V_{Q}$ to enable (22) and (23) to be used to estimate the relative amplitude and phase of the RF input voltage during the operation of the DF system. Details on how to determine the amplitude and DC offsets of $V_{I}$ and $V_{Q}$ can be found in [45].

## 5. System Measurements

After the bench tests of each system component, as discussed above, the integrated DF system was tested in an anechoic chamber to validate the proposed formulation. The experiments were carried out in the anechoic chamber of the Institute for Promotion and Industrial Coordination from the Brazilian Department of Aerospace Science and Technology. The DF system was tested under several conditions, exploiting the effects of changing the polarization of the incoming waves and the mutual coupling in the array. One RF source was employed at each experiment, and the DoA of the incoming waves was determined by applying a peak search routine to the values of the $S_{\mathrm{MUSIC}}$ function provided by the MUSIC algorithm to find the greatest peak. The experimental setup and equipment are summarized in Table 3.

### 5.1. Tests with the Printed Monopole Array

The printed monopole array was attached to the receiving circuit, as illustrated in Figure 14a, and this system was placed inside the anechoic chamber, as in the photo shown in Figure 14b. Since the array is composed of identical LP radiators, a pyramidal horn antenna of the same polarization was employed as a transmitting antenna. It is important to mention that, due to the array symmetry, an ambiguous solution for the DoA will occur because the corresponding $S_{\mathrm{MUSIC}}\left(\theta ,\phi \right)$ function gives the same value for both ϕ and 180°−ϕ angles, for every ϕ∈[−180°,180°]. Consequently, two peaks of equal amplitude are expected in the plot of $S_{\mathrm{MUSIC}}$ when a plane wave is incident on the array for a given direction (θ,ϕ).

In the experiment, the monopole array and the pyramidal horn are positioned at the same height, and the direction of propagation $\vec{k}$ of the plane waves radiated by the pyramidal horn is written using the coordinate system depicted in Figure 4a. Four different directions were considered: (1) $\vec{k}=-\hat{x}$ ($\theta_{w}=90{}^{\circ}$, $\phi_{w}=0{}^{\circ}$); (2) $\vec{k}=-\hat{y}$ ($\theta_{w}=90{}^{\circ}$, $\phi_{w}=90{}^{\circ}$); (3) $\vec{k}=-0.82\hat{x}-0.57\hat{y}$ ($\theta_{w}=90{}^{\circ}$, $\phi_{w}=35{}^{\circ}$); and (4) $\vec{k}=-0.26\hat{x}-0.97\hat{y}$ ($\theta_{w}=90{}^{\circ}$, $\phi_{w}=75{}^{\circ}$).

The normalized measured voltages $V_{S}^{-}$ at the comparator input for the four conditions listed above are shown in Table 4, which also presents the directions estimated by the implemented algorithm. Each voltage described in the table is the average value of M=200 samples. As seen, absolute errors less than 1° were verified in the four cases. In order to illustrate the MUSIC output, the plots of $S_{\mathrm{MUSIC}}$ for the four cases are illustrated in Figure 15. Note that for incoming waves arriving near or at the direction of the array axis (Cases 2 and 4), the peaks of the $S_{\mathrm{MUSIC}}$ function are not as well-defined as those exhibited if the waves arrive near or at broadside, showing a characteristic of the algorithm when used in DF systems comprised of linear arrays. Additionally, notice that to estimate the DoAs, no calibration of the DF system was required to compensate for the mutual coupling or the radiation patterns of the monopoles, for example.

### 5.2. Tests with the Microstrip Antenna Array

A photo of the microstrip antenna array attached to the receiving circuit is illustrated in Figure 16a, and the experimental setup placed inside the anechoic chamber is presented in Figure 16b. In this case, in which the array is composed of four rotated LP radiators, different transmitters were considered to exploit the property of polarization diversity. Additionally, note that no ambiguous solution is expected for the DoA estimation due to the asymmetric array geometry.

The coordinate system shown in Figure 6a was used in the experiment to describe the direction of propagation and the electric field of the incoming plane waves. Several combinations of directions and polarizations were tested, as detailed in Table 5. The polarizations of the selected transmitting antennas were horizontal LP (LPH), vertical LP (LPV), 45-degree LP (LPS), right-hand circularly polarized (RHCP), and left-hand circularly polarized (LHCP).

Table 5 presents the normalized measured voltages $V_{S}^{-}$ for the abovementioned conditions as well as the direction and polarization estimated by the implemented algorithm. The ratio q’ is evaluated at the direction $\theta^{\prime }=0{}^{\circ}$ and $\phi^{\prime }=0{}^{\circ}$, i.e., the direction of the line segment from the pyramidal horn to the positioner where the DF system was installed. Equation (19) was used to evaluate q’ from q and the rotation angles α and β that relate the coordinate systems attached to the array and to the anechoic chamber (which is classified as ground in the terminology of Section 3.2). It must be emphasized that the values of q and q’ match for Cases 4 and 5 since the incoming waves are circularly polarized in these tests.

In order to illustrate the MUSIC output, the plots of the $S_{\mathrm{MUSIC}}\left(\theta ,\phi \right)$ function are depicted in Figure 17. Indeed, there is only one well-defined peak at each tested case, whose level is at least 50 dB above the lower level exhibited by the $S_{\mathrm{MUSIC}}$ function, demonstrating that the DF system identified only one incoming wave. In addition, the DF system was able to scan 50° in elevation (θ) in all cases. In particular, even in Case 3, in which the incoming wave is near the boundary of the coverage region of the DF system, a sharp peak is observed, allowing for the estimation of the characteristics of the wave. Note also that the maximum absolute angular error is below 6°, and the estimated ratios q and q’ between the components of the incident electric fields are very close to the theoretical ones (third column in Table 5), which shows the validity of the formulation presented in this paper.

It is important to emphasize that the experiments were performed imposing different power levels in the transmitter. In these tests, the DoA was correctly estimated whenever the RF level at the comparator input was greater than −70 dBm. For each estimation, 200 samples (M) were collected.

As a way of illustrating the performance improvement obtained by using the formulation discussed in this paper, the MUSIC algorithm was run considering that the four antennas in the array of Figure 6 are isotropic radiators, i.e., all antenna patterns are equal and uniform [30]. Note that with this modification, the polarization of the incident waves can no longer be determined. The scattering matrices of the array and receiving circuit were the same as in the calculations for the microstrip antenna array. Figure 18 shows the plots of the $S_{\mathrm{MUSIC}}\left(\theta ,\phi \right)$ function for this new array, considering the normalized voltages of Cases 1 and 5 in Table 5. As seen, the MUSIC algorithm was not able to determine the correct DoA for both cases, and the plot of Case 1 exhibits ambiguity as well. Furthermore, the difference between the maximum and minimum values of the function is not greater than 12 dB, making it difficult to find the peaks.

## 6. Conclusions

This paper presented an alternative formulation of antenna arrays for DF systems that considers the active-element patterns of the array radiators as well as the scattering matrices of the array and receiving circuit. To the best of our knowledge, this is the first work to propose such a formulation to construct the transfer function of DF systems. As illustrated in the text, with the derived transfer function, the steering vectors of the DoA algorithms can be easily computed.

The active-element patterns and the scattering matrix of the array can be obtained using commercial electromagnetic simulators, and the scattering matrix of the receiver can be determined through bench tests with a VNA. Alternatively, the transfer function (7) also allows the use of active-element patterns measured in a near-field anechoic chamber, for example, and scattering parameters of the array measured with a VNA. As a result, unlike some publications found in the literature, our formulation does not require any type of calibration of the DF system to evaluate correction factors that account for mutual coupling and polarization mismatch effects. Moreover, it applies to any array geometry and receiving circuit architecture.

The differences between the radiation patterns of the antennas, the mutual coupling between them, the impedance mismatches, and couplings in the receiving circuit are all taken into account in the formulation, thereby contributing to the improvement of the accuracy of the output data and not restricting the use of arrays or receivers in which coupling is negligible. The transfer function (7) is also a useful resource to emulate highly coupled arrays and to assess the performance of the DoA algorithm under this condition. The paper still addressed the transformation of the DoA and the ratio between the components of the incident electric field, originally expressed in the array frame, to the coordinate system attached to the ground. The derived results are of special interest when the DF system is mounted on a moving platform.

A DF system operating at 2.2 GHz was designed and tested in an anechoic chamber to validate the developed formulation. The MUSIC algorithm was chosen to estimate the DoA and the polarization of the incoming plane waves. A description of the modules that comprise the system was given in detail. Two antenna arrays were used in the experiments: a linear array of quarter-wave monopoles, which is often found in DF systems, and a planar array of rectangular microstrip antennas exhibiting polarization diversity. As seen, good estimations were obtained, with angular errors less than 6°. In the case of the test with the microstrip antenna array, the polarization of the incident wave was also properly estimated. Currently, some adjustments are being made to the formulation to extend it to modulated signals (e.g., BPSK, QAM, etc.). Likewise, a study of the effect that the velocity of moving platforms has on the performance of our DF system prototype is ongoing. The relationship between the estimation errors and the platform velocity is being analyzed considering factors such as processing time and platform trajectory.

## Figures and Tables

**Figure 1 sensors-21-05048-f001:**
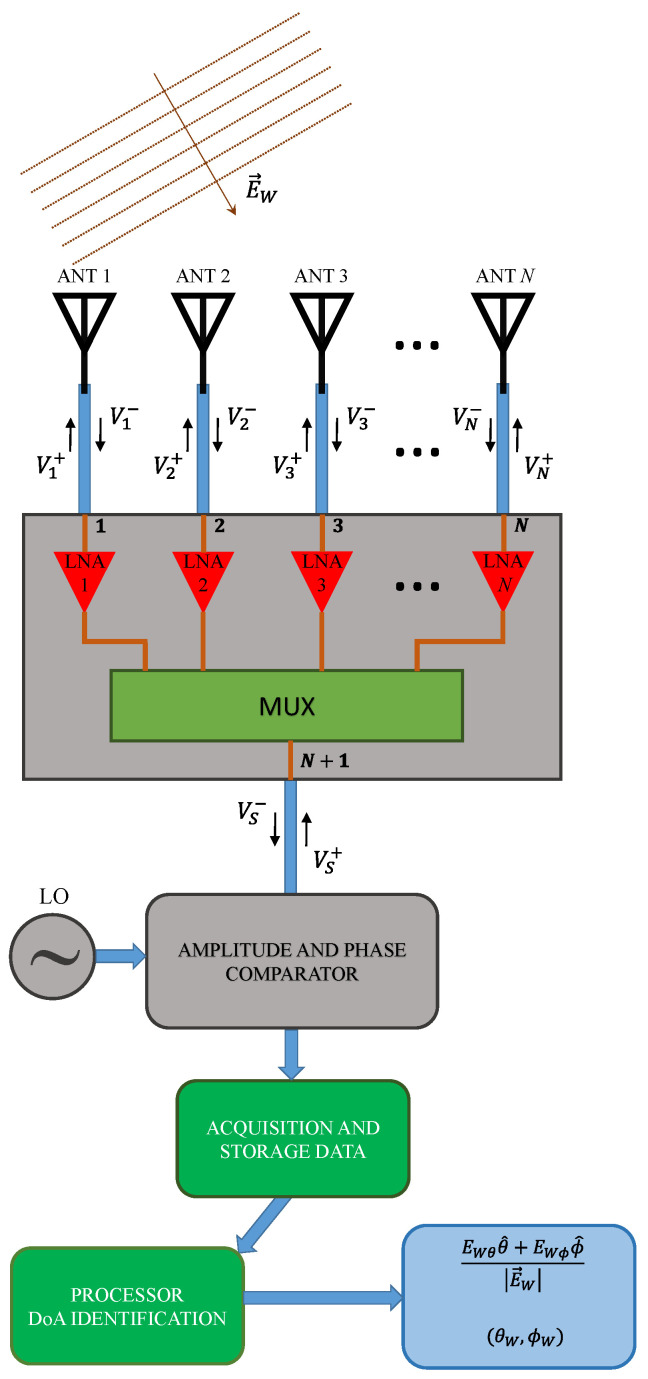
Direction finding system architecture.

**Figure 2 sensors-21-05048-f002:**
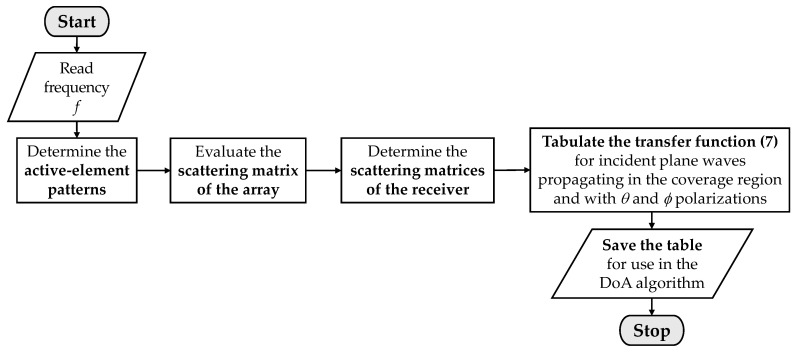
Flowchart of the formulation to compute the voltages used in the DoA algorithms.

**Figure 3 sensors-21-05048-f003:**
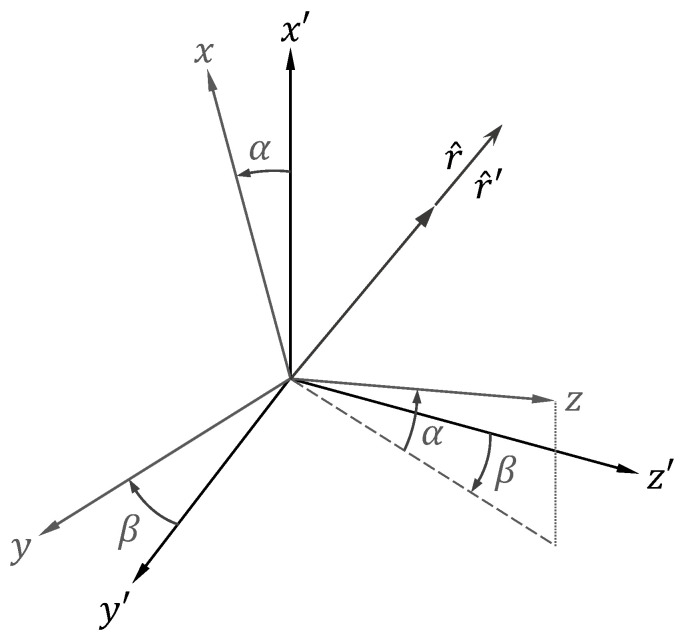
Coordinate systems attached to the antenna array (xyz) and to the ground (x’y’z’ ).

**Figure 4 sensors-21-05048-f004:**
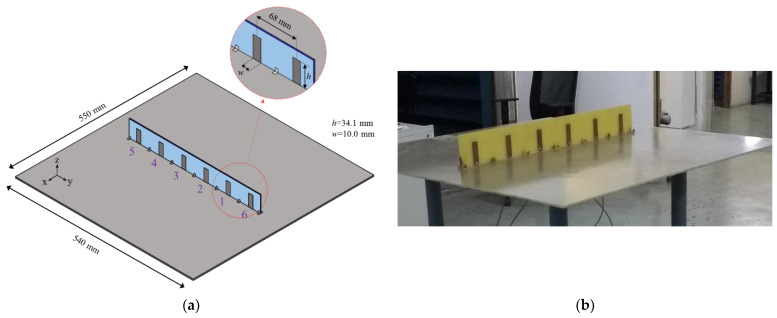
Printed monopole array: (**a**) Geometry dimensions; (**b**) Prototype.

**Figure 5 sensors-21-05048-f005:**
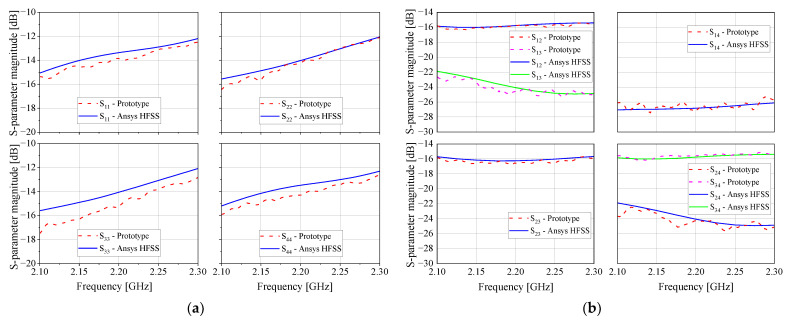
Simulated and measured S parameters of the printed monopole antenna array: (**a**) $S_{nn}$; (**b**) $S_{mn}$.

**Figure 6 sensors-21-05048-f006:**
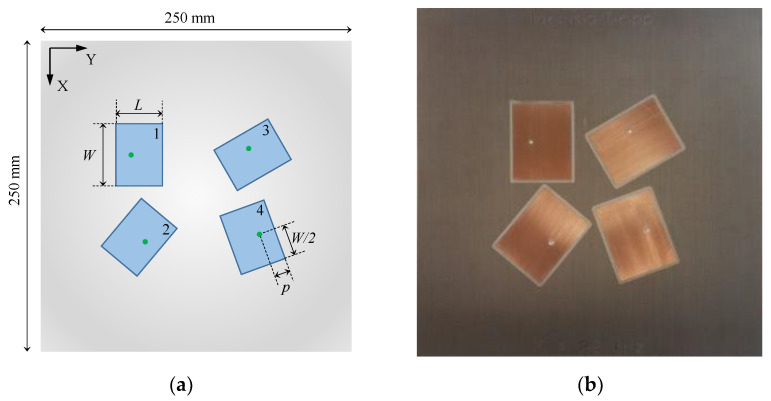
Designed microstrip antenna array: (**a**) Schematic; (**b**) Prototype.

**Figure 7 sensors-21-05048-f007:**
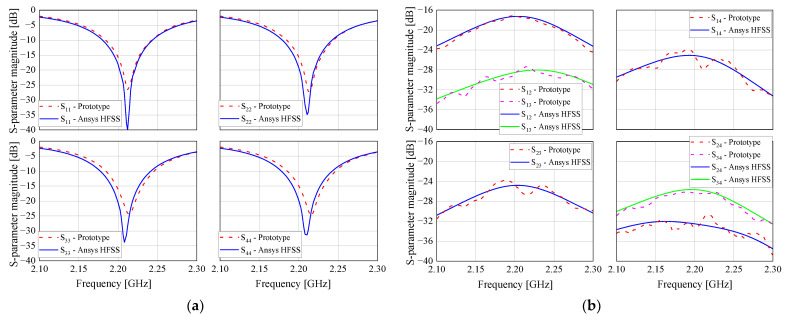
Simulated and measured S parameters of the microstrip antenna array: (**a**) $S_{nn}$; (**b**) $S_{mn}$.

**Figure 8 sensors-21-05048-f008:**
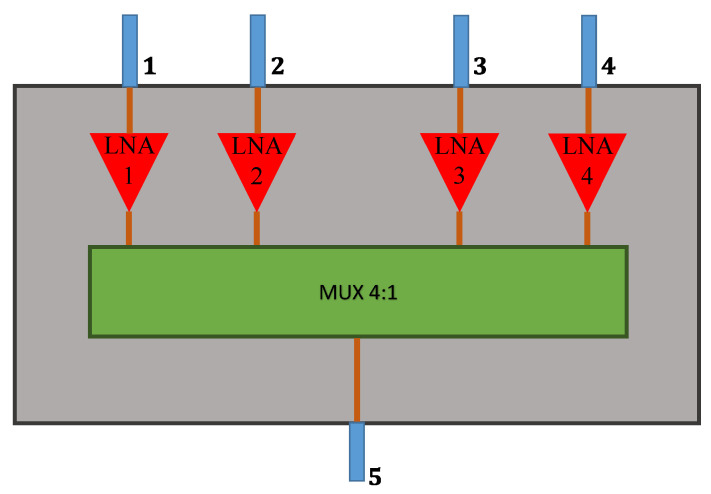
Schematic of the selection stage.

**Figure 9 sensors-21-05048-f009:**
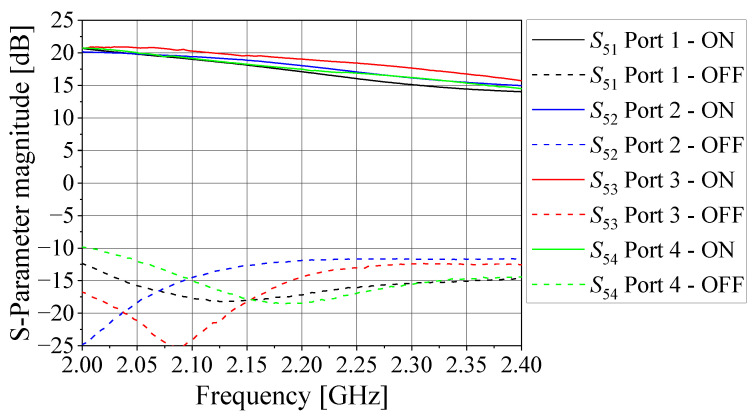
Magnitude of the transmission coefficients $S_{5n}$.

**Figure 10 sensors-21-05048-f010:**
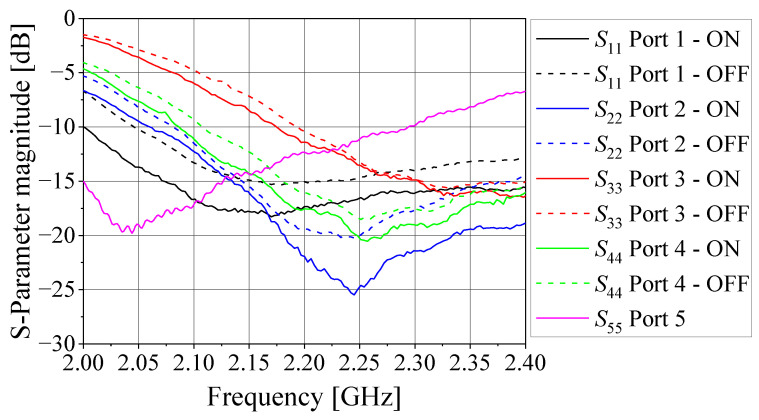
Magnitude of reflection coefficients $S_{jj}$.

**Figure 11 sensors-21-05048-f011:**
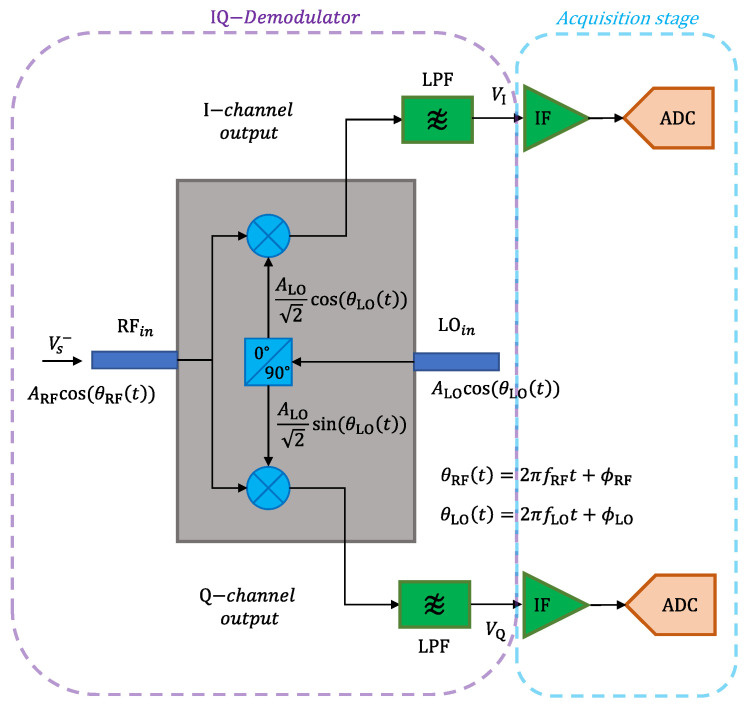
Schematic of the comparison and data acquisition stages.

**Figure 12 sensors-21-05048-f012:**
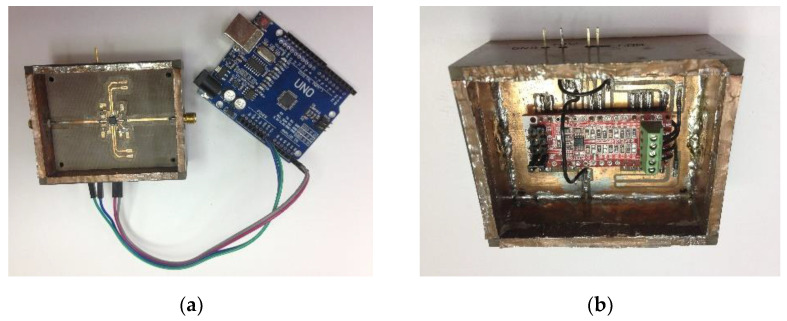
Comparison and acquisition circuit: (**a**) IQ-demodulator; (**b**) Analog-to-digital converter.

**Figure 13 sensors-21-05048-f013:**
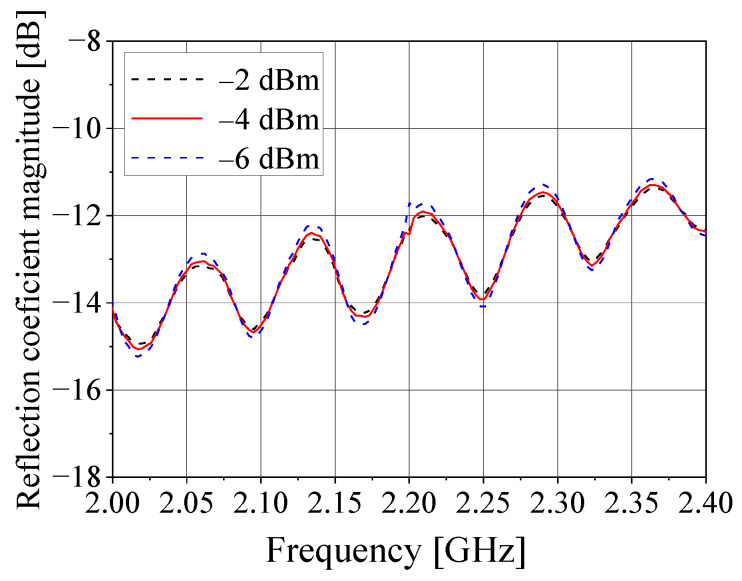
Reflection coefficient at the RF input port (RF*_in_*) of the comparison circuit.

**Figure 14 sensors-21-05048-f014:**
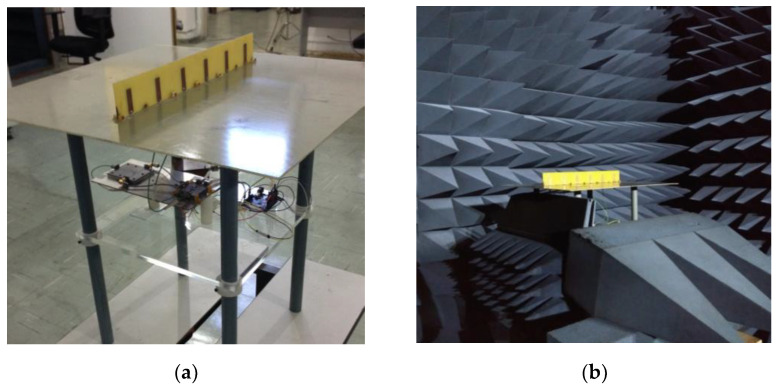
Printed monopole array: (**a**) Integrated with the receiving circuit; (**b**) Inside the anechoic chamber.

**Figure 15 sensors-21-05048-f015:**
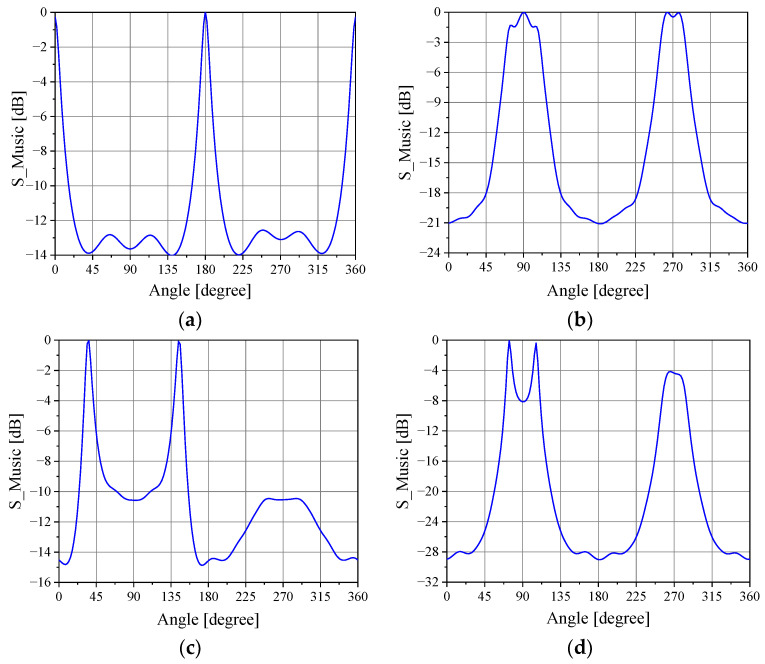
*S*_MUSIC_ plots: (**a**) Case 1; (**b**) Case 2; (**c**) Case 3; (**d**) Case 4.

**Figure 16 sensors-21-05048-f016:**
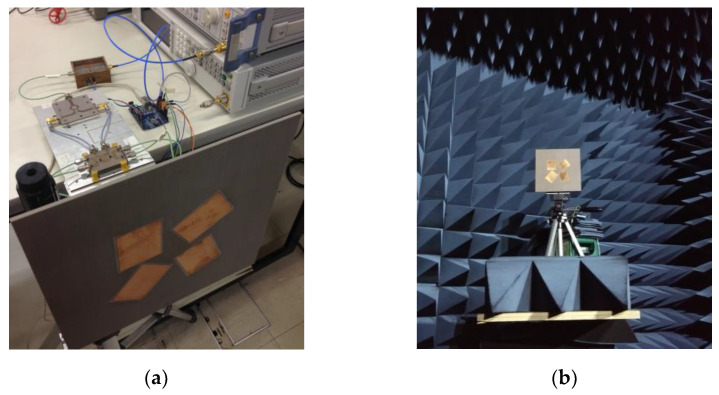
Printed monopole array: (**a**) Integrated with the receiving circuit; (**b**) Inside the anechoic chamber.

**Figure 17 sensors-21-05048-f017:**
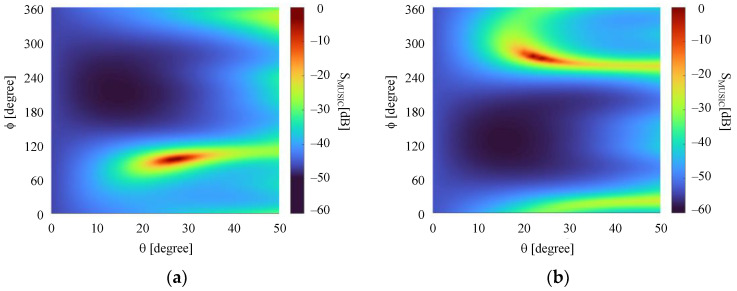
*S*_MUSIC_ plots: (**a**) Case 1; (**b**) Case 2; (**c**) Case 3; (**d**) Case 4; (**e**) Case 5; (**f**) Case 6; (**g**) Case 7.

**Figure 18 sensors-21-05048-f018:**
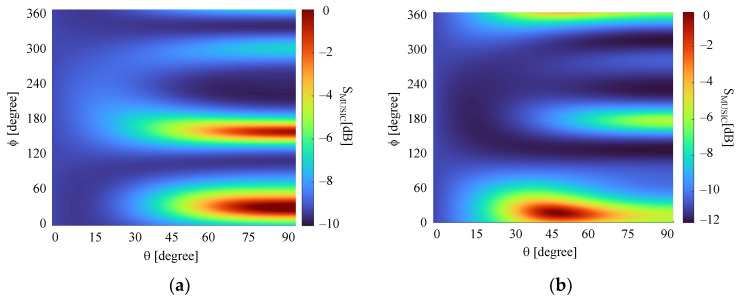
*S*_MUSIC_ plots for the array of four isotropic radiators: (**a**) Case 1; (**b**) Case 5.

**Table 1 sensors-21-05048-t001:** Comparison between the proposed formulation and other works.

Reference	Compute Mutual Coupling	Compute Radiation Pattern	Compute Receiving Circuit Transfer Function	Require Calibration	Polarization Estimation
This work	**✓**	**✓**	**✓**	**✕**	**✓**
[10,33]	**✓**	**✓**	**✕**	**✓**	**✕**
[11]	**✕**	**✓**	**✓**	**✓**	**✓**
[12,13,29,30,31]	**✓**	**✕**	**✕**	**✕**	**✕**
[15,16]	**✕**	**✕**	**✕**	**✓**	**✕**
[18,19]	**✕**	**✕**	**✕**	**✕**	**✕**
[20,21]	**✕**	**✓**	**✕**	**✓**	**✕**
[22,23]	**✓**	**✓**	**✕**	**✓**	**✕**
[24,25]	**✓**	**✓**	**✕**	**✕**	**✕**
[26,27,28,32]	**✕**	**✓**	**✕**	**✕**	**✓**

**Table 2 sensors-21-05048-t002:** Comparison between voltages computed with (3) and with Ansys HFSS.

Case	Angle of Incidence	Antenna Number	Normalized Voltages Ansys HFSS	Normalized Voltages Equation (3)
1	$\phi_{w}=30{}^{\circ}$	1	0.91∠−83°	0.91∠−83°
		2	0.99∠−171°	0.99∠−171°
		3	1.05∠95°	1.05∠94°
		4	0.94∠0.4°	0.94∠−0.3°
		5	0.88∠−78°	0.88∠−78°
		6	1∠0°	1∠0°
2	$\phi_{w}=60{}^{\circ}$	1	0.86∠−162°	0.88∠−163°
		2	0.81∠41°	0.82∠41°
		3	0.75∠−110°	0.75∠−111°
		4	0.75∠101°	0.74∠100°
		5	0.85∠−44°	0.84∠−45°
		6	1∠0°	1∠0°
3	$\phi_{w}=90{}^{\circ}$	1	1.01∠168°	1.01∠168°
		2	1.05∠−15°	1.05∠−15°
		3	1.12∠163°	1.12∠164°
		4	1.27∠−17°	1.27∠−17°
		5	1.58∠163°	1.56∠163°
		6	1∠0°	1∠0°

**Table 3 sensors-21-05048-t003:** Experimental setup and equipment.

Parameter/Equipment	Value/Model
Microwave generator	Rohde & Schwarz SMB100A
Transmitting antennas	ETS-Lindgren Double Ridged Horn Antenna 3115
	Circularly polarized patch antenna
Distance between transmitter and DF system	6 m
Frequency	2.2 GHz
Modulation	Continuous wave
Generator power	0 dBm

**Table 4 sensors-21-05048-t004:** Experimental results with the printed monopole array.

Case	Angle of Incidence	Antenna Number	Normalized Voltages	Estimated Direction
1	$\phi_{w}=0{}^{\circ}$	1	1∠0°	$\phi_{w}=0{}^{\circ}$ and $\phi_{w}=180{}^{\circ}$
		2	1.13∠9.1°	
		3	1.00∠−86.0°	
		4	1.03∠−75.9°	
2	$\phi_{w}=90{}^{\circ}$	1	1∠0°	$\phi_{w}=90{}^{\circ}$ and $\phi_{w}=264{}^{\circ}$
		2	1.20∠−163.5°	
		3	1.18∠99.8°	
		4	1.46∠−72.1°	
3	$\phi_{w}=35{}^{\circ}$	1	1∠0°	$\phi_{w}=36{}^{\circ}$ and $\phi_{w}=144{}^{\circ}$
		2	0.93∠124.7°	
		3	0.80∠−173.5°	
		4	1.03∠81.4°	
4	$\phi_{w}=75{}^{\circ}$	1	1∠0°	$\phi_{w}=74{}^{\circ}$ and $\phi_{w}=106{}^{\circ}$
		2	1.33∠−169.9°	
		3	1.20∠103.3°	
		4	1.25∠56.0°	

**Table 5 sensors-21-05048-t005:** Experimental results with the microstrip antenna array.

**Case**	Angles of Incidence	Pol. (Source)	Antenna Number	Normalized Voltages	Estimated Direction/Pol.
1	$\phi_{w}=90{}^{\circ}$	LPH q=0 $q{’}^{-1}=0$	1	1∠0°	$\phi_{w}=96{}^{\circ}$
	$\theta_{w}=25{}^{\circ}$		2	2.00∠−156.0°	$\theta_{w}=28{}^{\circ}$
	α=0°		3	1.11∠−154.6°	q=0.09+i0.02
	β=−25°		4	2.55∠25.0°	$q{’}^{-1}=0.01-i0.02$
2	$\phi_{w}=270{}^{\circ}$	LPV $q^{-1}=0$ q’=0	1	1∠0°	$\phi_{w}=270{}^{\circ}$
	$\theta_{w}=25{}^{\circ}$		2	0.16∠75.5°	$\theta_{w}=24{}^{\circ}$
	α=0°		3	1.10∠6.4°	$q^{-1}=0.03-i0.01$
	β=25°		4	0.51∠29.9°	$q^{\prime }=-0.03+i0.01$
3	$\phi_{w}=90{}^{\circ}$	LPS q=1 $q^{\prime }=-$1	1	1∠0°	$\phi_{w}=94{}^{\circ}$
	$\theta_{w}=50{}^{\circ}$		2	0.92∠−160.8°	$\theta_{w}=54{}^{\circ}$
	α=0°		3	0.89∠−108.6°	q=1.2−i0.1
	β=−50°		4	0.66∠−276.1°	$q^{\prime }=-0.90-i0.07$
4	$\phi_{w}=270{}^{\circ}$	RHCP q=i q’=i	1	1∠0°	$\phi_{w}=270{}^{\circ}$
	$\theta_{w}=40{}^{\circ}$		2	1.43∠−124.9°	$\theta_{w}=38{}^{\circ}$
	α=0°		3	0.91∠−48.9°	q=0.002+i0.96
	β=40°		4	0.97∠−126.4°	$q^{\prime }=-0.002+i1.04$
5	$\phi_{w}=90{}^{\circ}$	LHCP q=−i $q^{\prime }=-i$	1	1∠0°	$\phi_{w}=88{}^{\circ}$
	$\theta_{w}=20{}^{\circ}$		2	1.53∠−145.0°	$\theta_{w}=20{}^{\circ}$
	α=0°		3	1.59∠−178.6°	q=0.07−i0.91
	β=−20°		4	1.88∠−62.0°	$q^{\prime }=-0.09-i1.09$
6	$\phi_{w}=306.2{}^{\circ}$	LPH q=0.88 ${q^{\prime }}^{-1}=$0	1	1∠0°	$\phi_{w}=304{}^{\circ}$
	$\theta_{w}=38.3{}^{\circ}$		2	1.71∠122.2°	$\theta_{w}=40{}^{\circ}$
	α=−25°		3	0.65∠276.7°	q=0.88−i0.06
	β=30°		4	1.16∠184.4°	${q^{\prime }}^{-1}=0.00+i0.03$
7	$\phi_{w}=111.3{}^{\circ}$	LPH q=0.40 ${q^{\prime }}^{-1}=$0	1	1∠0°	$\phi_{w}=108{}^{\circ}$
	$\theta_{w}=32.5{}^{\circ}$		2	1.39∠−126.1°	$\theta_{w}=32{}^{\circ}$
	α=13°		3	0.86∠−117.7°	q=0.43−i0.05
	β=−30°		4	1.76∠36.1°	${q^{\prime }}^{-1}=-0.02+i0.04$

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
