# Peer review of "Alternative Formulation of Antenna Arrays for DF Systems Considering Active-Element Patterns and Scattering Matrices"

_sensors, 2021, doi:10.3390/s21155048_

Round 1
Reviewer 1 Report
This paper presented an alternative formulation of antenna arrays for DF systems that considers the active-element patterns of the array radiators as well as the scattering matrices of the array and receiving circuit. These works propose such a formulation to construct the transfer function of DF systems. With the derived transfer function, the steering vectors of the DoA algorithms can be easily computed.
The authors need to do the following revisions:
Add recent papers in the literature review and introduction i.e. published in the last three years.
There is no need to repeat the referenced papers again and again i.e. note line#41 and Line#42. you can put it only in line#42 in the second paragraph of the introduction section.
Figure 4 is not mentioned in the text of the paper. I do not understand the sifnificance of a figure without discussing it.
The authors need to add the limitation of the proposed scheme and future directions.
A table is needed with notations and their explanation.
Put a comma after "where" in line#311
Check for typos and spelling mistakes throughout the manuscript.
Author Response
Please,
Find attached the answers to your questions.
Best regards
Daniel Chagas do Nascimento

Reviewer 2 Report
This paper defines an alternative formulation of antenna arrays for DF systems that considers the active-element patterns of the array radiators /scattering matrices of the array and receiving circuit. The paper has potential however it requires a major revision w.r.t following points:
- SOA should be enriched with new literature in terms of table with advantages and disadvantages those methods have.
- Why MUSIC algorithm is proposed? Why not the DFT-based threshold system?
- What are the improvements that have been employed to the classical MUSIC algorithm?
- How is the complexity of this algorithm?
- A pseudocode of MUSIC algorithm emplyed with the system should be included.
- How does the Root locus diagram of the MUSIC algorithm looks like?
- For DF via MUSIC algorithm, is it tracking the first path of arrival or the maximum peak (strongest peak) of arrival?
- How does the paper compare w.r.t the following paper:
Estimation of Interference Arrival Direction Based on a Novel Space-Time Conversion MUSIC Algorithm for GNSS Receivers - STC-MUSIC algorithm enhances the purity of the noise subspace that improves the precision and robustness of the DOA estimation for interference signals significantly, how is the proposed work compared with STC-MUSIC algorithm?
- What type of channel environment is used? Is it AWGN or multipath such as TDL-A,B, and C? Are they 3GPP compliant?
- A table should be added summarizing experimental setup and parameters utilized
- For the case of moving platform, what is the speed? Is it from Factory automation or Indoor Factory Scenario?
Author Response

(The authors gave the same response as above.)

Reviewer 3 Report
This paper presents a technique for improving the accuracy of direction-finding (DF) systems by employing MUSIC and considering the effects of the antenna arrays, such as mutual coupling, received waves polarization, impedance mismatches. The correction is done by constructing the transfer function of the DF system using scattering matrices of the array. The authors provide the details of the analytical formulation combined with numerical results. Moreover, the proposed technique is experimentally verified using two types of receiving antenna arrays. The presented results sound solid with good accuracy of DoA detection.
However, I have a few comments regarding the manuscript.
First, in Figure 8 I think there is a typo in the legend.
Moreover, I do not understand why the authors changed the order of presenting two types of arrays, starting first with the microstrip antennas in section 4 and then starting with monopoles in section 5.
More importantly, I missed some performance comparisons with other techniques or without implementing the proposed one. As it is right now, I am not convinced that it improves the accuracy just because of the authors' claim. Moreover, although the proposed approach eliminates the need for calibrating the antenna arrays, it still requires scattering matrices. Please, correct me if I am wrong.
Finally, just as a curiosity, is it possible to reduce the ambiguity of the DoA detection using the monopoles by slightly modifying the array?
Author Response

(The authors gave the same response as above.)

Reviewer 4 Report
The authors present an Alternative Formulation of Antenna Arrays for DF Systems Considering Active-Element Patterns and Scattering Matrices. The paper is well written with an appropriate literature review pointing out the novelty proposed. The theorical background is is presented in detail, simulations and measurements were performed.
Some comments in order to improve the quality of the paper:
- In Section 2 it would be interesting to show a flowchart to facilitate the reader's understanding.
- Fig. 4 must be improved, it is very difficult to compare some simulation and measurement, especially for the antenna 1.
- The results (simulation and measurement) of the isolation between the antennas are requested.
- In 4.1.2. Printed Monopole Array, measurements are requested (Reflexion Coefficient and Isolation).
- Which Vector Network Analyzer (VNA) was used for the measurements?
- Lines 467 - 468, Fig 11 show the reflexion coefficient (negative) and not the return loss (positive).
More explanations regarding Fig. 15 are expected.
Author Response

(The authors gave the same response as above.)
